# Lithium Occurrences in Brines from two German Salt Deposits (Upper Permian) and First Results of Leaching Experiments

**Michael Mertineit and Michael Schramm ***

Federal Institute for Geosciences and Natural Resources (BGR), Stilleweg 2, 30655 Hannover, Germany;
michael.mertineit@bgr.de

\* Correspondence: michael.schramm@bgr.de

**Abstract:** Lithium occurrences were detected in Upper Permian (Zechstein) salt rocks and saline solutions of the Gorleben and Morsleben salt structures, northern Germany. The brine occurrences were mainly connected to anhydrite rock-bearing formations and to lithological boundaries. Most of these brines display a high Mg content and were accordingly interpreted as intrasalinar solutions, which developed during sedimentation, diagenesis, and the subsequent rock–fluid interaction. These Mg-rich brines frequently show high Li concentrations. One of the assumptions made, is that Li was leached from phyllosilicates, since no natural Li-bearing salt minerals are known to date. To improve the understanding of the origin of Li in the brines, leaching experiments were performed on the Li-bearing phyllosilicate Lepidolite. Lepidolite with a Li content of 2.42 wt. % served as an analogue material, which was exposed to 18 saline solutions of different composition for a period of three years. The most pronounced leaching effect (53.36 μg Li/g in the brine) was observed during the interaction with a 0.03 mol/kg $H_2O$ $MgCl_2$ solution, the second most pronounced by modern seawater interaction. The experiments show that the amount of Li leached from the lepidolite is dependent on brine composition.

**Keywords:** Permian phyllosilicate bearing strata; brines; leaching experiments; lepidolite; lithium

## 1. Introduction

An important topic in salt research are geochemical characteristics of brines, their host rocks, and the interaction processes between salt rocks and solutions. For this reason, the origin and genesis of brines in Permian Zechstein salt deposits have been the focus of many studies, e.g., by References [1–4]. For example, occurrences of saline solutions were detected in all North German salt mines [5]. Most of these solutions have been classified as relicts of Permian seawater, which were trapped and stored within the salt during sedimentation and subsequent processes [6]. Due to their potentially hazardous influence on mining activities, they are handled with high priority. In this context, potential migration paths, genesis, as well as the origin of the brines, are of interest.

Apart from saline solutions, gas and hydrocarbon occurrences are also observed in salt deposits [5,7,8]. However, the present study focuses on brines and their geochemical properties.

Various geochemical signatures can be used for genetic interpretations of salt minerals and brines [9], the most important are the trace elements Br and Rb. Therefore, substantial knowledge exists about the origin, the distribution in brines and minerals, and the thermodynamic properties of these elements [3,10–17]. However, due to locally occurring brines with high Li concentrations in salt deposits, Li gets increasing attention in salt research.

High contents of Li (up to 7000 ppm, [8]) are found and currently mined in different salt deposits in South America, e.g., the salt deposits of the Salar de Atacama, Chile, the Salar de Hombre Muerto,

Argentinia [8,18] and the Salar de Uyuni, Bolivia [8,19,20]. The main source of Li is related to water–rock interactions with volcanic country rock (Bolivia) [21]. These brines result from evaporation. Li concentrations of some oil field brines (e.g., Smackover Formation, Gulf Coast (TX and FL, USA [22,23]) may reach >100 mg/L [23]. However, apart from these general observations, only little knowledge exists about the principle geochemical behaviour of Li in evaporites, especially in relation to the interaction between brines and minerals.

In contrast to highly concentrated Li brines in salt deposits, there is no evidence of naturally occurring Li salts or hints of significant Li contents in naturally formed salt minerals. The occurrence of Li-carnallite, interpreted to have formed in salt lakes in South America, motivated the first experimental and crystallographic studies [24,25]. The authors of Reference [26] published a probable $Li_2SO_4$ formation in Salar de Uyuni. In order to examine the behaviour of Li in this system, the authors of Reference [26] evaporated these brines for up to 54 days. They found that during the first 34 days, the Li and $SO_4$ concentrations in the brine increased, but from the 35th day, the concentration in the solution decreased, which was interpreted as a result of $LiSO_4$ precipitation (indirect proof).

Currently, it is unknown if the detection of very low quantities of Li in the lower ppm range originate from fluid inclusions, or whether Li is incorporated in the crystal lattice of naturally formed salt minerals.

In brines and rocks of the Upper Permian salt deposits of the Gorleben salt dome and the Morsleben salt structure, both located in the Southern Permian Basin, northern Germany (Figure 1), high Li concentrations of up to 401 µg/g in brines [27] and 161 µg/g in bulk rock samples [28] were measured. The Gorleben salt dome consists of Upper Permian (Zechstein) rock salt formations. The salt dome is aligned in the NE–SW direction and is ca. 14 km long. The salt table is located ca. 250 m below ground level [5]. The salt movement started in the Early Triassic and during Upper Jurassic periods to the Lower Cretaceous period, and the salt rocks penetrated the overburden and created a salt dome. The final stages of salt rise occurred during the Upper Cretaceous and Paleogen periods [29,30]. The Gorleben salt dome was investigated for its suitability to construct a repository for high-level radioactive waste between 1979–2000 and 2010–2012. In the Gorleben exploration mine, saline solutions were collected continuously at different sites between ca. 1996 and 2012, all of the solutions originating from anhydrite rocks [5,31].

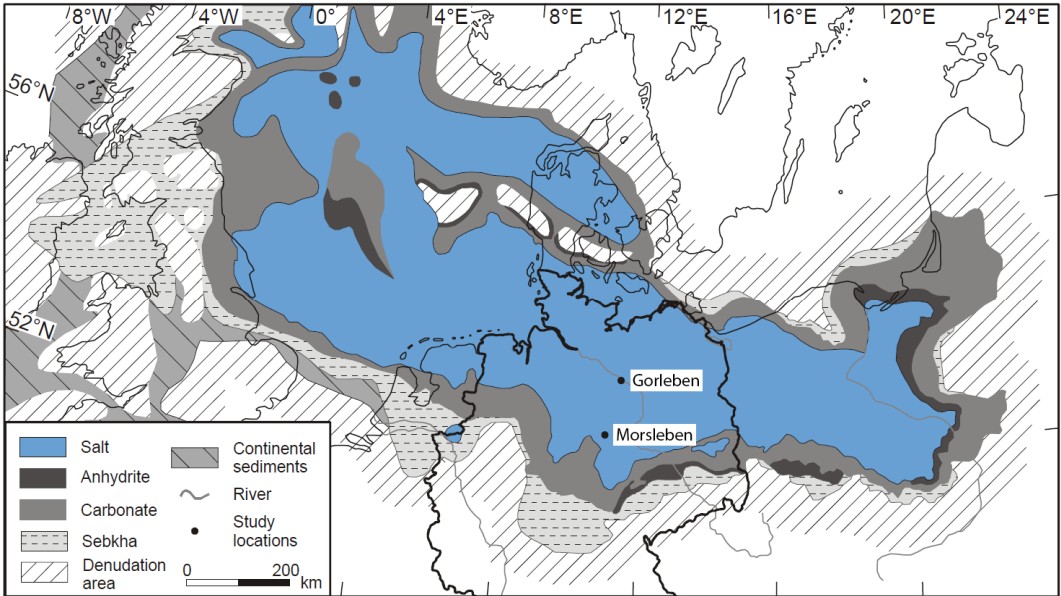

**Figure 1.** Distribution of the Zechstein rocks (Upper Permian) in Northern Europe showing the investigated salt deposits of Gorleben and Morsleben, (map from Reference [32]; modified).

The Morsleben salt structure is located in the northeastern part of the Subherzynian basin, at the southern rim of the Zechstein basin. In this region, Zechstein salt migrated into the NW–SE trending Allertal fault zone [33] and underwent various types of deformation. The main salt migration took place from the Upper Triassic to Cretaceous periods with material inflow mainly from the west to the east [34]. The salt body is primarily regarded as a tectonic structure and not a halokinetic one [34]. In the Morsleben mine, brine samples of two influxes have been collected and analysed sporadically since 1907, and continuously since 1991 [35].

References [10,36] already linked elevated Li concentrations in brines to be originated from phyllosilicate-bearing strata (also see References [8,37]). Typical lithostratigraphic units of this type are, for example, the Grauer Salzton (z3GT), Leinekarbonat (z3LK), Hauptanhydrit (z3HA), Tonmittelsalz (z3TM), Roter Salzton (z4RT) and the Tonbrockensalz (z4TS) (Figure 2).

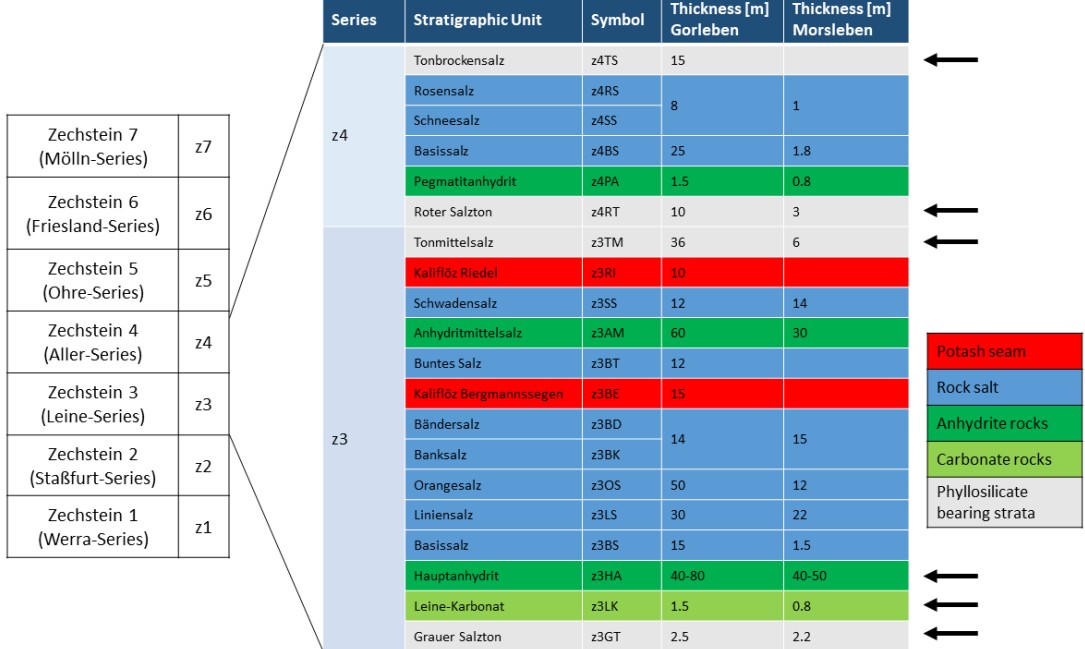

**Figure 2.** Simplified stratigraphic scheme of the Zechstein (Upper Permian) in the North German Zechstein Basin from Reference [5], modified. Thicknesses of the stratigraphic units according to Reference [5] (Gorleben) and References [34,38] (Morsleben). Arrows display stratigraphic units containing significant amounts of possible Li-sources in phyllosilicate-bearing strata.

The "Hauptanhydrit" (z3HA) is an anhydrite rock unit, with a thickness of ca. 40–80 m. The unit is subdivided into 13 zones, distinguishable by differences in composition, sedimentary-diagenetic structures, grain size and thickness. The base of the Hauptanhydrit is characterized by ca. 4 wt. % carbonate (mainly magnesite, minor dolomite and calcite) and traces of quartz and mica. In certain areas, the magnesite content can increase to 22 wt. %, caused by algal layers [5,39]. At the top of the layer, the magnesite content decreases to ca. 2 wt. % [39].

The footwall of the Hauptanhydrit consists of the "Grauer Salzton" (z3GT), a maximum of 2.5 m thick phyllosilicate-bearing rock and the "Leine-Karbonat" (z3LK), a carbonate rock of maximum 1.5 m thickness. The composition of these stratigraphic units differs, depending on the sedimentary conditions and the position within the Zechstein basin due to the transport distance of clastic material from the backcountry.

In Gorleben, in the centre of the basin, the composition is more homogeneous. The main components of the z3GT are anhydrite (ca. 55 wt. %), hydrotalcite, chlorite and quartz, and the minor components are magnesite, halite, illite and kaolinite [5]. The z3LK consists of magnesite and anhydrite, only trace amounts of hydrotalcite were observed [5].

In Morsleben, at the rim of the basin, the main components of the z3GT are quartz, muscovite-illite (ca. 30 wt. % each), with additional trace amounts of tourmaline, chlorite, serpentine and halite [39]. The z3LK consists of magnesite (ca. 56 wt. %), anhydrite (ca. 20 wt. %) and minor amounts of quartz, muscovite-illite, koenenite, chlorite and calcite [39].

The transition between the Leine- and the Aller-Series is characterised by the Tonmittelsalz (z3TM) and the Roter Salzton (z4RT). Close to the top of the Aller-Series, the Tonbockensalz (z4TS) is developed. These units are characterized by a high amount of halite of ca. 80–90 wt. % [39], with minor amounts of quartz, anhydrite, muscovite, chlorite-smectite, kaolinite and magnesite [28]. Clay rock occurrences are interrupted by intercalations of anhydrite and halite layers. The z4TS could not be observed in the Morsleben salt structure.

Organic matter is contained in all stratigraphic units described, often enriched in thin layers associated with carbonate and quartz/phyllosilicates.

Depending on the lithology of the samples, Li concentrations are highly variable [28]. Potentially Li-bearing minerals within phyllosilicate-containing strata are muscovite and chlorites.

Due to relatively low Li concentrations in seawater (0.17 µg/g; [40]) and evaporated seawater, from which the salts originate, it is unclear where the high Li concentrations derive from. The Li concentrations of brines detected in salt deposits are in the range of bulk rock analyses of Upper Permian (Zechstein) phyllosilicate-bearing strata [28], both of which are some magnitudes higher than the Li concentrations of the most strongly evaporated seawater. Lepidolite was used to investigate leaching effects between brines and phyllosilicates because lepidolite is comparable with muscovites and chlorites that are common components of the phyllosilicate-bearing strata. Another advantage of using lepidolite is its very high Li content, advantageous for getting a measurable leaching effect in a manageable timescale for laboratory experiments. Experimental studies on the leaching behaviour were performed by exposing lepidolite to 18 saline solutions of different composition for the duration of ca. three years.

## 2. Analytical Methods and Experimental Setup

### 2.1. Sampling of Natural Brines

In the Gorleben salt dome, the inflow of solutions was linked to mining activity. During excavation of galleries, at certain points, connected to changes in lithology and lithostratigraphic boundaries, saline solutions entered the mine. However, most solution influxes had a limited volume of few $dm^3$ to several $m^3$ [5]. Brine that originated from more extensive and long-lasting solution influx were collected by a system of tailraces, tubes and accumulation bins. The gathered brine was sampled on a regular basis.

In the Morsleben mine, a similar construction is used to collect brine at mining claim 1A. The volume of inflowing brine is ca. 1.4 $m^3$/a [35]. The influx rate has been sporadically analysed since 1962, and systematic analyses of the composition and trace element content have been carried out in monthly intervals since 1991. At the second solution influx, mining claim H, the brine inflow started in 1907 due to extensive mining activities. Consequently, a protection embankment was built, thus the exact position of the influx is not clear. Furthermore, the volume is higher (ca. 10 $m^3$/a, [35]), therefore the brine is collected in a pool. The influx rate has been monitored since 1907, and systematic analyses of the composition and trace element content have also been carried out since 1991.

### 2.2. Experimental Setup

For each experimental run, 8 g of lepidolite (from Minas Gerais, Brazil; stoichiometric formula: $K(Li,Al)_{2-3}((OH,F)_2/Si_3AlO_{10}))$ with a lithium concentration of 2.42 wt. % (for more details see Section 3.2.1) was ground to a grain size <200 µm and added to 100 g solution (mass ratio $1_{(rock)}:12.5_{(solution)}$). The compositions of the solutions vary from double distilled $H_2O$, NaCl, KCl, $MgCl_2$ solutions, modern seawater to artificial solutions (Table 1), approximately comparable to the

composition of the so-called solutions Q, R and Z (according to References [10,41]). These solutions are basically halite, anhydrite and partly polyhalite saturated. Solution Q is saturated with respect to sylvite, carnallite and kainite. Solution R is saturated with respect to kieserite, carnallite and kainite. Solution Z is saturated with respect to kieserite, carnallite and bischofite (Figure 3). With the exception of seawater and double distilled $H_2O$, the solutions were prepared using pure NaCl (Emsure ACS, ISO for analysis, Merck), KCl (pro analysi, Merck), $MgCl_2 \cdot 6H_2O$ (extra pure for table water, Merck) and $MgSO_4 \cdot H_2O$ (Sigma-Aldrich). The first solution used for the experiments was double distilled $H_2O$, representing the largest difference in concentration with an electrical conductivity of 0.055 μS/cm (sample 1).

Three pure NaCl solutions were used for the experiments:

- 0.42 mol NaCl/kg $H_2O$ is a typical NaCl-content of fresh seawater (sample 2);
- 4.96 mol NaCl/kg $H_2O$ assigns first halite precipitation from evaporating seawater (sample 3);
- 5.74 mol NaCl/kg $H_2O$ is close to the theoretical halite saturation at 6.11 mol/kg $H_2O$ (sample 4) in a pure NaCl solution.

Six pure artificial KCl solutions were created:

- 0.01 mol KCl/kg $H_2O$ represents the KCl content of fresh seawater (sample 5);
- 0.02 mol KCl/kg $H_2O$ almost corresponds to solution Z at the point of bischofite formation at the end of seawater evaporation (sample 6);
- 0.19 mol KCl/kg $H_2O$ is typical at halite formation during seawater evaporation (sample 7);
- 0.37 mol KCl/kg $H_2O$ assigns KCl concentration of evaporating seawater at polyhalite saturation (sample 8);
- 0.60 mol KCl/kg $H_2O$ represents almost solution Q, equilibrium with sylvite, carnallite and kainite at 25 °C (sample 9), and
- 4.29 mol KCl/kg $H_2O$ at sylvite saturation in a pure KCl solution (sample 10).

Four pure $MgCl_2$ solutions were used:

- 0.03 mol $MgCl_2$/kg $H_2O$, representative for fresh seawater (sample 11);
- 3.50 mol $MgCl_2$/kg $H_2O$, as in solution Q at 25 °C (sample 12);
- 4.26 mol $MgCl_2$/kg $H_2O$, as in solution R at 25 °C (sample 13), and
- 5.51 mol $MgCl_2$/kg $H_2O$ near bischofite saturation, almost representing solution Z at 25 °C (sample 14).

Sample 15 represents seawater from the North Sea. Additionally, three artificial solutions that correspond approximately to the solutions at the invariant points are Q (sample 16), R (sample 17) and Z (sample 18). All three solutions are comparable to the natural analogue with the exception of the trace components (Br, Rb, Li and Si).

The lepidolite samples in the solutions were shaken at a temperature of 22 °C to 25 °C on a shaking table with a frequency of 150 shakes/min. During the first year, the process was interrupted only shortly after 24 to 26 days for measuring the electrical conductivity. After one year, the solutions were separated from the lepidolite by filtration with Sartorius membrane filters (pore size <0.45 μm and <0.1 μm, Sartorius AG, Göttingen, Germany), washed with double distilled $H_2O$, cleaned with ethanol and dried at room temperature. In a second step, half of the reacted lepidolite and half of the reaction solution were merged again (in the same rock-water-mass-relation of $1_{(rock)}:12.5_{(solution)}$), and the experiments continued for two more years following the same conditions described above. The entire experiment lasted for ca. three years.

**Table 1.** Main ($Na^+$, $K^+$, $Ca^{2+}$, $Mg^{2+}$, $Cl^-$, $SO_4^{2-}$ in wt. %) and trace ($Si^{4+}$, $Li^+$, $Rb^+$, $Cs^+$ in µg/g) components of the initial solutions (numbers 1 to 18), the solutions after one year (numbers 19 to 36) and three years of reaction (numbers 37 to 54). Empty cells indicate no analyses performed, - = below detection limit, DI $H_2O$ = double distilled water, sw = seawater, Si content of seawater in italics according to Reference [40].

| No. | 1 | 2 | 3 | 4 | 5 | 6 | 7 | 8 | 9 | 10 | 11 | 12 | 13 | 14 | 15 | 16 | 17 | 18 |
|---|---|---|---|---|---|---|---|---|---|---|---|---|---|---|---|---|---|---|
| N sample | H2O/ | N/2/i | N/3/i | N/4/i | K/5/i | K/6/i | K/7/i | K/8/i | K/9/i | K/10/i | M/11/i | M/12/i | M/13/i | M/14/i | S/15/i | Q/16/i | R/17/i | Z/18/i |
| type of solution | DI H2O | NaCl solutions | | | | KCl solutions | | | | | | MgCl2 solutions | | | | sw | solution Q | solution R | solution Z |
| pH | 7.0 | 5.6 | 6.1 | 5.9 | 5.8 | 5.6 | 5.5 | 5.5 | 5.5 | 5.5 | 5.7 | 4.1 | 4.0 | 3.9 | 7.9 | 5.2 | 4.9 | 4.3 |
| density (g/cm³) | 0.996 | 1.016 | 1.170 | 1.193 | 0.997 | 0.998 | 1.006 | 1.015 | 1.026 | 1.171 | 0.999 | 1.235 | 1.270 | 1.332 | 1.020 | 1.289 | 1.307 | 1.338 |
| Na+ | - | 0.948 | 8.853 | 9.890 | - | - | 0.005 | 0.005 | 0.004 | 0.013 | - | 0.019 | 0.018 | 0.016 | 0.936 | 0.715 | 0.338 | 0.100 |
| K+ | - | - | 0.001 | 0.002 | 0.042 | 0.083 | 0.743 | 1.424 | 2.243 | 12.707 | - | - | 0.018 | 0.015 | 0.036 | 1.585 | 0.410 | 0.063 |
| Ca2+ | - | - | - | - | - | - | - | - | - | - | - | - | - | - | 0.035 | 0.001 | 0.001 | 0.001 |
| Mg2+ | - | - | - | 0.001 | - | - | - | - | - | - | 0.069 | 6.390 | 7.407 | 8.869 | 0.113 | 6.290 | 7.189 | 8.312 |
| Cl- | - | 1.450 | 13.511 | 15.155 | 0.037 | 0.074 | 0.668 | 1.285 | 2.104 | 11.447 | 0.203 | 18.223 | 21.090 | 24.904 | 1.667 | 19.736 | 21.337 | 25.186 |
| SO4 2- | - | - | - | - | - | - | - | - | - | - | - | 0.005 | 0.005 | 0.003 | 0.226 | 1.761 | 2.158 | 0.349 |
| Si4+ | - | - | - | - | - | - | - | - | - | - | - | - | - | - | *2.81* | - | - | - |
| Li+ | - | - | - | - | - | - | - | - | - | - | - | - | - | - | 0.166 | - | - | - |
| Rb+ | - | 0.027 | 0.073 | 0.077 | 0.011 | 0.025 | 0.654 | 0.325 | 0.501 | 2.715 | 0.002 | 0.016 | 0.009 | 0.008 | 0.119 | 0.352 | 0.084 | 0.018 |
| Cs+ | - | 0.002 | 0.007 | 0.007 | - | - | 0.035 | 0.002 | 0.002 | 0.006 | - | 0.001 | 0.001 | 0.001 | 0.002 | 0.001 | 0.001 | - |

| No. | 19 | 20 | 21 | 22 | 23 | 24 | 25 | 26 | 27 | 28 | 29 | 30 | 31 | 32 | 33 | 34 | 35 | 36 |
|---|---|---|---|---|---|---|---|---|---|---|---|---|---|---|---|---|---|---|
| N sample | H2O/1 | N/2/1 | N/3/1 | N/4/1 | K/5/1 | K/6/1 | K/7/1 | K/8/1 | K/9/1 | K/10/1 | M/11/1 | M/12/1 | M/13/1 | M/14/1 | S/15/1 | Q/16/1 | R/17/1 | Z/18/1 |
| pH | 8.5 | 7.9 | 7.3 | 6.3 | 8.1 | 7.9 | 7.9 | 7.7 | 7.7 | 7.5 | 7.2 | 4.5 | 4.1 | 3.5 | 7.8 | 5.3 | 4.6 | 3.9 |
| density (g/cm³) | 0.997 | 1.017 | 1.171 | 1.194 | 0.998 | 0.998 | 1.006 | 1.015 | 1.026 | 1.170 | 0.999 | 1.236 | 1.271 | 1.331 | 1.020 | 1.288 | 1.307 | 1.337 |
| Na+ | 0.005 | 1.053 | 8.717 | 9.707 | 0.006 | 0.006 | 0.011 | 0.011 | 0.010 | 0.017 | 0.006 | 0.015 | 0.013 | 0.012 | 0.970 | 0.721 | 0.332 | 0.097 |
| K+ | 0.006 | 0.023 | 0.030 | 0.036 | 0.041 | 0.081 | 0.731 | 1.406 | 2.281 | 12.737 | 0.014 | 0.083 | 0.035 | 0.031 | 0.068 | 1.624 | 0.412 | 0.055 |
| Ca2+ | - | - | - | - | - | - | - | - | - | - | - | 0.001 | 0.001 | 0.001 | 0.030 | 0.001 | 0.001 | 0.001 |
| Mg2+ | - | - | - | - | - | - | - | - | - | - | 0.064 | 6.533 | 6.947 | 8.847 | 0.122 | 6.329 | 7.179 | 8.341 |
| Cl- | 0.009 | 1.652 | 13.395 | 15.056 | 0.046 | 0.084 | 0.675 | 1.296 | 2.159 | 11.550 | 0.215 | 18.872 | 21.261 | 25.662 | 1.840 | 20.000 | 21.253 | 25.328 |
| SO4 2- | 0.001 | 0.001 | 0.001 | 0.001 | - | - | - | 0.001 | 0.001 | - | - | 0.001 | 0.001 | 0.001 | 0.228 | 1.793 | 2.167 | 0.308 |
| Si4+ | 15.942 | 7.814 | 1.596 | 1.566 | 12.182 | 11.242 | 7.434 | 5.989 | 5.465 | 1.997 | 16.372 | 2.647 | 1.838 | 1.053 | 9.163 | 1.815 | 1.431 | 1.049 |
| Li+ | 32.701 | 41.396 | 43.114 | 45.397 | 33.378 | 33.871 | 37.376 | 39.030 | 37.802 | 38.285 | 36.225 | 36.643 | 37.362 | 35.383 | 39.106 | 37.422 | 37.500 | 37.551 |
| Rb+ | 4.331 | 17.242 | 29.229 | 31.677 | 15.441 | 21.943 | 37.917 | 38.795 | 36.874 | 39.690 | 11.733 | 29.039 | 28.782 | 27.880 | 22.048 | 31.313 | 21.954 | 19.549 |
| Cs+ | 0.328 | 1.526 | 3.590 | 3.979 | 1.302 | 2.007 | 4.657 | 5.177 | 5.132 | 5.809 | 0.851 | 3.914 | 4.093 | 4.097 | 1.833 | 5.241 | 4.463 | 4.107 |

| No. | 37 | 38 | 39 | 40 | 41 | 42 | 43 | 44 | 45 | 46 | 47 | 48 | 49 | 50 | 51 | 52 | 53 | 54 |
|---|---|---|---|---|---|---|---|---|---|---|---|---|---|---|---|---|---|---|
| N sample | H2O/3 | N/2/3 | N/3/3 | N/4/3 | K/5/3 | K/6/3 | K/7/3 | K/8/3 | K/9/3 | K/10/3 | M/11/3 | M/12/3 | M/13/3 | M/14/3 | S/15/3 | Q/16/3 | R/17/3 | Z/18/3 |
| pH | 7.7 | 7.4 | 6.2 | 5.9 | 7.5 | 7.4 | 7.4 | 7.3 | 7.3 | 6.8 | 6.8 | 4.2 | 3.9 | 3.4 | 7.1 | 4.1 | 3.8 | 3.3 |
| density (g/cm³) | 0.996 | 1.017 | 1.171 | 1.195 | 0.997 | 0.998 | 1.006 | 1.015 | 1.027 | 1.171 | 0.999 | 1.236 | 1.270 | 1.327 | 1.020 | 1.288 | 1.305 | 1.333 |
| Na+ | | 1.110 | 9.129 | 10.298 | 0.006 | 0.007 | 0.005 | 0.004 | 0.009 | 0.014 | 0.007 | 0.011 | 0.008 | 0.010 | 0.956 | 0.694 | 0.341 | 0.059 |
| K+ | | 0.020 | 0.026 | 0.025 | 0.044 | 0.084 | 0.784 | 1.539 | 2.379 | 13.506 | 0.014 | 0.039 | 0.025 | 0.021 | 0.049 | 1.612 | 0.402 | 0.051 |
| Ca2+ | | - | 0.001 | 0.002 | - | - | - | - | 0.001 | 0.001 | - | 0.001 | 0.001 | 0.001 | 0.031 | 0.002 | 0.001 | 0.001 |
| Mg2+ | | - | - | 0.005 | - | - | - | - | 0.055 | - | 0.066 | 6.920 | 7.717 | 8.758 | 0.113 | 6.547 | 7.689 | 8.805 |
| Cl- | | 1.760 | 13.912 | 16.086 | 0.051 | 0.090 | 0.730 | 1.407 | 2.296 | 12.287 | 0.227 | 19.997 | 22.444 | 26.159 | 1.783 | 20.388 | 22.141 | 26.314 |
| SO4 2- | | 0.001 | 0.001 | - | 0.001 | - | 0.001 | 0.001 | 0.001 | 0.009 | 0.001 | 0.001 | 0.001 | 0.001 | 0.247 | 1.930 | 2.351 | 0.393 |
| Si4+ | | 14.753 | 12.805 | 10.878 | 21.063 | 19.047 | 12.923 | 12.809 | 11.688 | 25.609 | 29.033 | 25.075 | 18.107 | 31.640 | 13.727 | 20.967 | 13.030 | 72.795 |
| Li+ | | 45.144 | 43.963 | 46.440 | 41.424 | 41.301 | 43.939 | 45.423 | 44.219 | 38.841 | 53.361 | 33.163 | 40.151 | 36.913 | 50.593 | 40.381 | 39.857 | 39.775 |
| Rb+ | | 17.210 | 42.934 | 48.970 | 18.963 | 27.096 | 52.444 | 58.054 | 61.877 | 84.939 | 10.698 | 46.470 | 53.059 | 59.486 | 25.707 | 54.205 | 46.455 | 48.727 |
| Cs+ | | 1.221 | 4.822 | 5.876 | 1.494 | 2.429 | 6.520 | 7.902 | 8.696 | 13.654 | 0.599 | 6.512 | 7.846 | 9.405 | 1.933 | 10.773 | 9.533 | 9.679 |

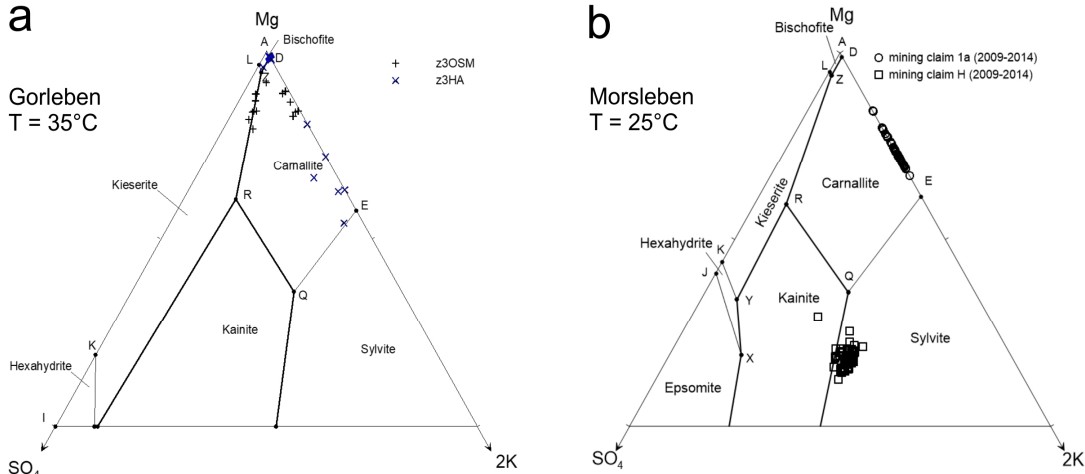

**Figure 3.** Geochemical character of the investigated brines, displayed in the upper part of the Jänecke diagram (80–100% Mg, 0–20% K, 0–20% SO$_4$; modified according to Reference [41]. (**a**) Brines from anhydrite rock-bearing strata of the Gorleben site, where the maximum Li concentrations were detected. + = Gorleben-Bank (z3OSM), x = main anhydrite-bearing formation (z3HA), *T* = 35 °C. (**b**) Morsleben brines at *T* = 25 °C from mining claims 1a and H.

## 2.3. Analytical Methods

The initial, unaltered lepidolite was analysed using XRD (PANalytical MPD Pro, Malvern Panalytical GmbH, Kassel, Germany), XRF (PANalytical Axios) using the total fusion (tetra-) borate flux method, ICP-OES (Agilent Technologies 5100, LabWrench, Santa Clara, CA, USA) and ICP-MS (Thermo Fisher iCAP Q, Thermo Fisher Scientific, Waltham, MA, USA). The solutions, initial and after reaction, were analysed with respect to density (Anton PAAR DMA 38, Anton Paar GmbH, Graz, Austria), electrical conductivity (WTW Multi 3420, and WTW TetraCon 925, Xylem Analytics Germany Sales GmbH & Co. KG, WTW, Weilheim, Germany), pH (WTW MultiLine P4 and WTW SenTix 81) as well as main, minor (ICP-OES, Spectro Arcos, Spectro Analytical instruments GmbH, Kleve, Germany) and trace components (ICP-MS, Agilent 7500/7700, LabWrench, Santa Clara, CA, USA). Furthermore, both initial and altered lepidolite were analysed by optical microscopy and SEM (FEI Quanta 650 MLP, Thermo Fisher Scientific, Waltham, MA, USA) analyses.

The standard deviation (2$\sigma$) for the ICP-OES Spectro Arcos is <1% for Na, K, Ca, Mg, Cl and SO$_4$ of the analysed brines. The dissolved lepidolite was analysed using ICP-OES (Agilent Technologies 5100) with a RSD (relative standard deviation) <1% for Al, Ca, K, Li, Mg, SO$_4$ and <1.5% for Na. The ICP-MS (Thermo Fisher iCAP Q) shows a standard deviation of (2$\sigma$) < 2%.

Analyses of rock samples and natural brines from the Gorleben and Morsleben mines occurred the same way [28,35], with certain differences of the used hardware due to the long duration of the monitoring.

Modelling of the mineral saturation of the natural brines and the seawater was performed with the software package EQ3/6v7.2 (version 7.2, Lawrence Livermore National Laboratory, Livermore, CA, USA) [42] special version c [16,17], using the thermodynamic data base hmw (version R10, Lawrence Livermore National Laboratory, Livermore, CA, USA) [43], modified according to [16,17].

## 3. Results

### 3.1. Li-Occurrences in the Salt Deposits of Gorleben and Morsleben

All solutions in contact with rock salt detected in salt mines are typically saturated with respect to halite. The composition and chemical saturation of the solutions, except for Ca-sulfate minerals, can be displayed in ternary Jänecke diagrams according to Reference [41] (Figure 3). Due to the depth of –820 to –930 mBSL (meter below sea level) at the Gorleben site and –167 to –231 mBSL at the Morsleben site,

the diagrams in Figure 3 are calculated and created for different temperatures ($T = 35$ °C and $T = 25$ °C, respectively), representing local geological conditions.

The geochemical character of the brines in the anhydrite rock-bearing strata of the Gorleben salt deposit, where the maximum concentrations of Li in brines were detected, are displayed in Figure 3a. The brines are saturated with respect to halite and, mainly, carnallite. Some brines are additionally saturated with respect to kieserite, or in some cases, bischofite. The pH values vary between 2.0 and 6.1.

The geochemical character of brines occurring in the Morsleben site (mining claim 1a and H) are illustrated in Figure 3b. All investigated solutions of the Morsleben site are saturated with respect to halite. The composition of the solutions from mining claim H, sampled between 2009 and 2014, show saturation with respect to sylvite and in parts, with respect to kainite. The brines from mining claim 1a are saturated with respect to carnallite, controlled by EQ3/6 modelling. The pH values vary between 3.4 and 5.8 in the brines of solution access mining claim H.

The Li concentrations of the Gorleben and Morsleben brines vary between 0.24 µg/g and 401 µg/g (Appendix A, Tables A1, A3 and A4) [27,35].

The Li concentrations of Morsleben groundwater vary between 0.008 µg/g and 0.684 µg/g (Appendix A, Table A2).

### 3.2. Experimental Investigations

### 3.2.1. Lepidolite

Before starting the experiments, the sample material was tested for purity, with respect to lepidolite. X-ray diffraction (XRD) analyses show that the sample material consists of lepidolite and probably trilitionite. Polarizing microscopy of the ground material showed acicular lepidolite (Figure 4a) with some additional aggregates of very fine-grained lepidolite (Figure 4b). The geochemical analyses of the initial lepidolite yield 22.74 wt. % Si (XRF), 8.38 wt. % K, 14.30 wt. % Al, 2.42 wt. % Li (ICP-OES), 1.54 wt. % Rb, 0.32 wt. % Cs (ICP-MS) and F (not quantified). Small Cs-rich domains in lepidolite were detected using SEM-EDS (Quantax EDS for SEM, Bruker Corporation, Billerica, MA, USA) (Figure 4c).

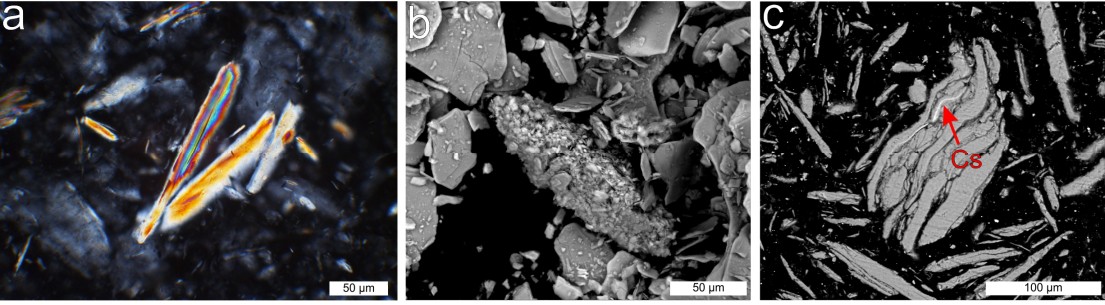

**Figure 4.** Microphotographs of initial lepidolite. (**a**) Microphotograph of scattered lepidolite in ground material, + polarizers. (**b**) SEM photograph of lepidolite aggregate. (**c**) SEM photograph of lepidolite with a small Cs-rich domain.

The altered material did not differ significantly from the initial lepidolite with respect to mineralogy. The overall grain size and occurrence of aggregates remained the same (Figure 5a,b) with the main difference being locally developed cements, infilling the space between imbricated lepidolite crystals (Figure 5c). The cement consists of very fine-grained lepidolite (grain size <10 µm), which probably formed due to mechanical grinding during the shaking process.

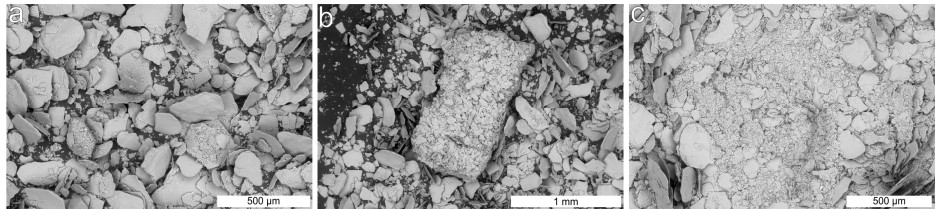

**Figure 5.** SEM photographs of altered lepidolite. (**a**) Overview of lepidolite. (**b**) Lepidolite aggregate. (**c**) Very fine-grained lepidolite cement between imbricated lepidolite crystals.

### 3.2.2. Solutions—Starting Solutions and Reaction Solutions after One and Three Years

Table 1 shows the density, the pH and the composition of the starting solutions (upper part of Table 1) and the reaction solutions (one year of reaction; middle part of Table 1 and three years of reaction; bottom part of Table 1).

The reactant (lepidolite) as well as the interacting solutions were analysed after one year, and again at the end of experiments after three years. During the first year, the electrical conductivity was measured every 24–26 days, in order to monitor the continuation of the reaction progress [44].

For the estimation of the reaction progress and the specimen distributions in the reacting solutions, thermodynamic modelling using EQ3/6 was performed. For Si, Cs, Rb and Li, no thermodynamical data of satisfying quality have been implemented in the thermodynamic database of EQ3/6 to date. Therefore, EQ3/6 modelling was performed only for Na, K, Ca, Mg, Cl and $SO_4$ using the thermodynamic database hmw [43].

All reaction solutions with the exception of double distilled $H_2O$ are low to very high saline solutions. The degree of dissolved components, i.e., salinity, corresponds to the electrical conductivity: rising conductivity documents an increasing quantity of dissolved components representing an ongoing reaction progress.

The measurements of the density show no significant differences between initial and reacting solutions.

The geochemical analyses document changes in the pH values and the composition/concentration of the resulting solutions due to interaction reactions between solutions and lepidolites (Figure 6). In all cases, the solutions show enrichments of Li, Si, Rb and Cs, independent from their initial geochemical composition. K is leached only by double distilled $H_2O$, NaCl and low concentration $MgCl_2$ solution.

The pH values of the investigated experimental saline solutions changed in comparison to the initial pH of the solutions. After three years, the pH of the reaction solutions varied between 3.3 and 8.5, depending on the kind of solution and reaction time (Table 1, Figure 6). The most significant change in the pH was observed for the KCl solutions: from the initial 5.5 to 5.8 to a pH in the reaction solutions of 7.5 to 8.1 after one year and a pH of 6.8 to 7.5 after three years. The initial pH of the 0.42 molal NaCl solution shifts from 5.6 to a pH of 7.9 in the reaction solution (after one year) and to a pH of 7.4 (after three years). The pH of the 0.03 molal $MgCl_2$ solution increased from 5.7 to a pH value of 7.2 in the reaction solution (after one year) and 6.8 after three years. Remarkable is the little decrease in the pH value of these solutions reacting after one year in comparison to the solutions reacting after three years (Figure 6).

After a one-year duration of the experiments, the maximum enrichment with a content of 45 μg/g of Li was detected in the 5.74 molal NaCl solution (sample 4–22). After three years, the highest Li concentration with 53 μg/g was shown by the solution with the lowest $MgCl_2$ concentration of 0.03 mol/kg $H_2O$ (sample 11–47). The maximum of reaction progress between one and three years of reaction was also observed in the $MgCl_2$ solution with the lowest concentration, the second most reaction progress was detected in the modern seawater interaction solution (samples 15–33–51), followed by lower concentrated K solutions (samples 5–23–41, 6–24–42). The results, especially related to the massive reaction increase between one and three years, suggest that the interaction reactions between lepidolite and these solutions are not finished after three years (Figure 6).

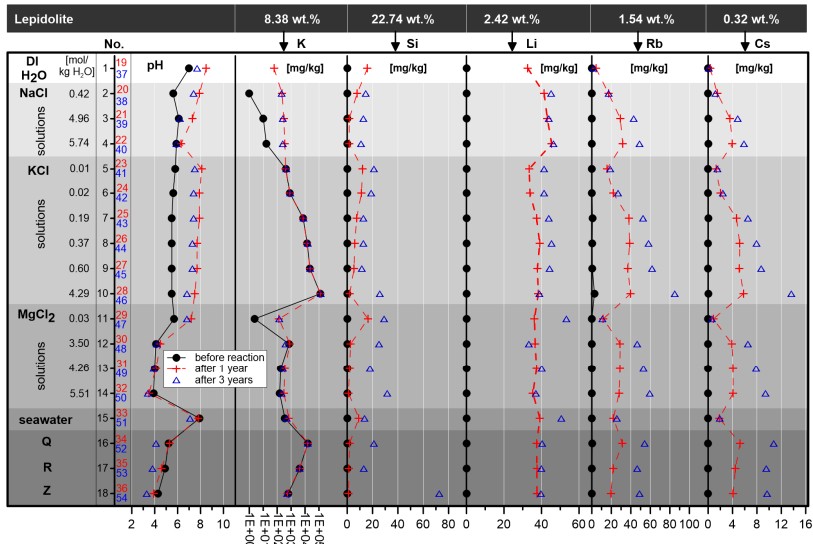

**Figure 6.** K, Si, Li, Rb and Cs concentrations and pH of initial solutions, after one year (red) and after three years of reaction (blue). The K content is displayed in logarithmic scale; DI $H_2O$ = double distilled $H_2O$. The K, Si, Li, Rb and Cs content of the initial lepidolite is displayed for comparison.

The Si content in the reacting solution increased after one year in seawater up to 16.37 µg/g and after three years, the highest Si concentration, with 72.79 µg/g, was detected in solution Z (reaction very likely not finished).

The highest Rb content in the reaction solution was detected in the KCl solution with the highest concentration. The Rb value increased from 39.69 µg/g (sample 28) after one year, to 84.93 µg/g (sample 46) after three years of reaction.

The highest Cs content was also detected in the KCl solution with the highest concentration (sample 10), representing an increase from a maximum of 5.80 µg/g after one year to 13.65 µg/g after three years of reaction.

Potassium is leached by double distilled $H_2O$, NaCl (Figure 6), $MgCl_2$ solutions (with the exception of sample 12–30, 48) and some potassium solutions that reacted for three years (samples 7, 8, 9,10–43, 44, 45, 46; Table 1, not noticeable in Figure 5 due to logarithmic scale). The largest release from the lepidolite was detected with 0.799 wt. % K in sample 46. Virtually no difference in the dynamic of leaching in relation to K between one and three years was observed with the exception of the higher concentrated K solutions (Table 1).

## 4. Discussion

### 4.1. Natural Occurrences

Published data for Li concentrations in seawater vary between 0.166 µg/g [45], 0.17 µg/g [40] and 0.176 µg/g [46]. These data are in good agreement with analyses of surface seawater from the North Sea, which show a mean value of 0.167 µg/g. During evaporation of seawater, the Li content in the residual brine increases continuously (Figure 7), because the precipitating evaporites neither form Li minerals nor incorporate Li in the minerals in detectable concentrations. Theoretical considerations and experimental investigations of seawater evaporation suggest that the Li concentration rises in the residual solution from 0.17 µg/g to 1.37 µg/g when halite is first precipitated. The Li concentration in the solution when the first polyhalite is formed is 4.57 µg/g, at the first kieserite precipitation 11.49 µg/g and at the first kainite crystallization 13.25 µg/g (Figure 7, Table 2, Appendix A, Table A5).

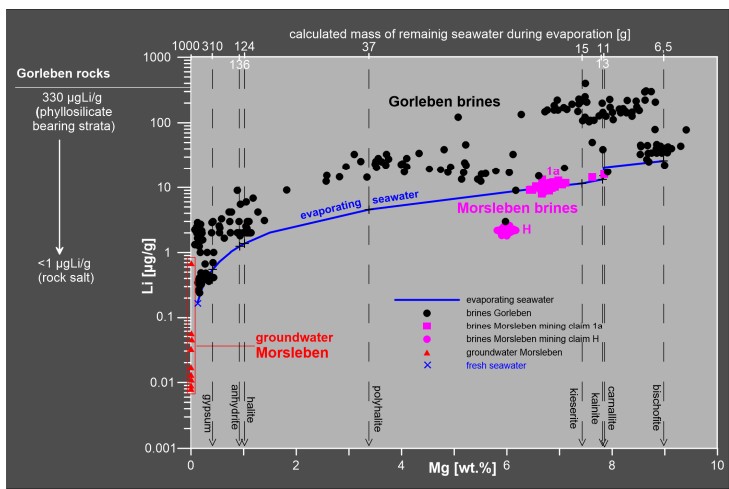

**Figure 7.** Li and Mg concentrations of brines from Gorleben and Morsleben. For comparison, the Li content of the groundwater-monitoring network from Morsleben and the Li content of the rocks from Gorleben are displayed. In addition, the development of the Li content in evaporating seawater (blue line) and the first precipitates from seawater are shown.

The Mg concentration in modern seawater according to Reference [40] is 0.1278 wt. %. The development of the Mg concentration during evaporation of seawater is described in Figure 7.

In the Gorleben site, brines and some exemplarily selected rocks (phyllosilicate-bearing strata from the z3TM and z4TS) as well as brines from the Morsleben site have been investigated geochemically. Depending on stratigraphic unit and lithology, the mineralogical composition and hence, the Li concentrations of the rocks, vary between <1 µg/g (typical rock salt) and 330 µg/g (phyllosilicate-bearing strata, z3GT, deep drilling Go1005). The phyllosilicate-containing strata are composed of halite, anhydrite, quartz, muscovite and chlorite. We assume that these phyllosilicates are Li carriers, but alternative Li sources, like fluid inclusions with metamorphic brines, could be possible. In addition, organic compounds may play a role in elevating Li concentrations. Examples in this context were observed in source rocks of oil field brines in the Gulf of Mexico and Alberta [23].

Most of the Gorleben brines are enriched with respect to Li, and brines with high Mg contents show higher Li concentrations (Figure 7). The highest Li concentration (up to 401 µg/g) has been detected in a brine with a high Mg content of about 7.5 wt. % (Figure 7). The Li values are significantly higher than in seawater of the highest evaporation stage (26 µgLi/g). Therefore, it is assumed that the investigated solutions are generated from the interaction with phyllosilicates containing Li. A relationship between Li occurrences in brines and Li-bearing phyllosilicates in clay- and salt-rich strata, detected in several salt deposits in Germany, had already been suggested by References [3,10]. The Li-enriched solutions in both salt deposits are interpreted as internal metamorphic brines originating from potash seams, as fluids migrated through fissures and porous areas in phyllosilicate-bearing strata during diagenesis and the salt structure evolution. Li represents an additional geochemical signature.

The majority of the solution occurrences observed in the Gorleben exploration mine are halite-saturated, and some very highly concentrated K-Mg brines. However, some samples are solutions of low concentrations with respect to Mg and Li and are unsaturated with respect to halite. These are fresh water-derived (e.g., groundwater) and infiltrate the exploration mine through the shafts (external solutions; Figure 7). Other halite under saturated brines are interpreted to be technical solutions, caused by underground road construction and mine ventilation.

In the Morsleben salt mine, two solution accesses are currently active. The brines of mining claim 1a plot on or a little bit above the evaporation line of the Li/Mg ratio (Figure 7), representing probably highly evaporated relictic Permian seawater, or metamorphic brines of internal origin demonstrated by the Rb/Br ratios [35]. The brines of mining claim H plot below the evaporation trend line (Li/Mg), indicating fresh water which interacted with salt minerals (also proven by Rb/Br ratios in Reference [35]).

References [47,48] found Li concentrations in brines from Zechstein salt rocks in northern Germany that are comparable to our data. The Li content was between 140 µg/g to 490 µg/g [47] and 460 µg/g [48] and correlated with a high bitumen content of maximum 2870 µg/g in the investigated rocks [48]. The high Li concentrations were accordingly interpreted as formation solutions, which were trapped and stored in salt rocks and were in contact with bituminous material.

**Table 2.** $K^+$ concentration (wt. %) and $Si^{4+}$, $Li^+$, $Rb^+$ and $Cs^+$ concentration (µg/g) in seawater during evaporation and precipitation of salt minerals. *according to Reference [49], **according to Reference [50], *** = main precipitation period of carnallite, n.r. = no reaction between precipitates and solution, w.r. = with reaction between precipitates and solution.

| Type of Solution | Seawater | | | | Evaporating Seawater | | | | |
|---|---|---|---|---|---|---|---|---|---|
| Precipitates (EQ3/6 calculation) | | Gypsum | Halite | Polyhalite | First Precipitates of Kieserite | Kainite | Carnallite | | Bischofite |
| Evaporation factor of first occurrence | 1 | 3.22 | 8.07 | 26.87 | 67.58 | 77.94 | 88.44 | ***121.58 | 153.64 |
| $K^+$ | 0.0399 | 0.1287 | 0.3220 | 10.718 | 0.3152 | 0.4599 | 0.4578 | 0.4577 | 0.0538 |
| $Si^{4+}$ | 2.81 | 9.67 | 24.21 | 80.61 | 202.75 | 233.83 | 265.32 | 364.76 | 460.92 |
| $Li^+$ | 0.17 | 0.55 | 1.37 | 4.57 | 11.49 | 13.25 | 15.04 | 20.67 | 26.12 |
| $Rb^+$ | 0.12 | 0.38 | 0.96 | 3.22 | | *5.84 | *7.86 | *8.11 | **0.25 n.r. **0.01 w.r. **0.23 |
| $Cs^+$ | 0.0003 | 0.0009 | 0.0023 | 0.0078 | 0.0197 | 0.0227 | 0.0258 | 0.0355 | 0.0449 |

Concerning other Li occurrences worldwide, the Li concentrations of the brines in Gorleben and Morsleben are comparable to the maximum Li concentration of brines, e.g., documented for the Salton Sea (California and Mexico) [23]. In contrast, the brines of the Great Salt Lake in Utah (maximum 43 mg/L and 64 mg/L), the brines of Searles Lake in California (80 mg/L) and the brines of Clayton Valley in Nevada (maximum 300 mg/L) show much lower concentrations [23], whereas higher Li concentrations are documented for the brines of the Salar de Atacama in Chile (up to 7000 mg/L), which represents the highest Li content. Thus, it is obvious that the variations in the Li content of the Upper Permian (Zechstein) brines are less pronounced compared to the Li concentrations of the brines from continental deposits. Due to the diversity of different continental source rocks (e.g., evaporites, lake sediments, volcanic rocks, pegmatites, clays), the solutions that originated and interacted with these rocks are different in composition, which also affects the Li content [23].

*4.2. Experiments*

The experimental approach represents, with the set P-T-conditions (atmospheric pressure and ca. 25 °C), the sedimentary to early diagenetic situation. During the experiments, obvious changes in the geochemical composition of the reaction solutions were detected. Depending on the reaction time, the geochemical composition and the pH of the reaction solutions, different components of the lepidolite in varying quantities were leached (Figure 6). The experimental investigations indicate Li enrichments in any solution during leaching of lepidolite, as well as, for example, Rb, Cs and Si. Moreover, obvious changes in the pH were observed. In contrast, no changes in the density of the reaction solutions were detected, which could be related to small quantities of material, transformed and leached during the reaction process.

In contrast to the $MgCl_2$-free solutions, the NaCl und KCl reaction solutions become more alkaline with up to 2.0–2.5 units higher than in the initial solutions. In the $MgCl_2$-free solutions, the lepidolite shows a buffering effect for KCl and parts of NaCl solutions (Figure 6). It is assumed that OH groups are released from the crystal lattice of the lepidolite to a certain degree. For example, the NaCl solution sample 2 shows a pH value of 5.6 and an $OH^-$ content of $6.07 \times 10^{-9}$ mol/kg $H_2O$. After one year of reaction, the pH increased to a value of 7.9 (sample 20) and an $OH^-$ content of $1.23 \times 10^{-6}$ mol/kg $H_2O$. The balance between both solutions, regarding $OH^-$ referred to 100 g solution, is $2.08 \times 10^{-6}$ wt. % ($2.08 \times 10^{-4}$ mg). Since 8 g of lepidolite, with an OH content of about 4.5 wt. % (360 mg OH) (according to the stoichiometric formula), reacted in 100 mL solution, only very little OH of the lepidolite is necessary

to change the pH. The balance of $2.08 \times 10^{-4}$ mg in the solution represents only $5.79 \times 10^{-5}$ wt. % of the OH of the lepidolite. An important process could be the reaction of NaCl to NaOH in the NaCl solutions and from KCl to KOH in the KCl solutions.

In contrast to the 0.03 to 4.26 molal $MgCl_2$ solutions, a very weak decrease in the pH was detected in the 5.51 molal (mol/kg $H_2O$) $MgCl_2$ solution (sample 14) and seawater (sample 15). A more pronounced decrease in the pH was measured in the solutions Q and R (samples 12 and 13), up to a pH of 3.3. As one of the results of the lepidolite corrosion, for example, Si was released into the reaction solution, which would affect the crystal lattice. However, microstructural investigations concerning these effects have not been performed yet.

Changes in the K concentrations between the initial solutions and those after reaction were observed in double distilled $H_2O$, the NaCl and in most of the $MgCl_2$ solutions (with the exception of the 3.50 molal $MgCl_2$ solution). The two highest concentrated KCl solutions (initially 0.60 and 4.29 molal KCl), showed a K release from the mineral into the solution in comparable low quantities in the first year. After three years, the release increased, also in the 0.37 molal KCl solution. The KCl solutions with an initial KCl content of 0.01 and 0.02 mol/kg $H_2O$ showed no significant change. In comparison, the development of the K content in the evaporating seawater at different evaporation levels (indicated by the first occurrence of typical salt minerals, based on EQ3/6 [41] modelling), is displayed in Table 2. Additional information is given in terms of the Li, Rb, Si and Cs concentrations.

The strongest effect of Li leaching was observed in the interaction reaction with the Mg-bearing solution with the lowest concentration (0.069 wt. % Mg; sample 11) and with modern seawater (0.113 wt. % Mg, 0.036 wt. % K; sample 15). It is notable that leaching is more effective in solutions with lower concentrations compared to solutions with higher concentrations with respect to Mg. In addition, especially the K solutions with lower concentrations are still actively leaching, although not as effectively as the $MgCl_2$ solutions. Depending on the coordination number (CN) 6 in the crystal, the ionic radius of $Mg^{2+}$ (0.72 Å [46]) is comparable with the ionic radius of $Li^+$ (0.76 Å [46]), which might be the reason for a certain exchange, certainly persisting a charge difference. Because in the lepidolite both $Li^+$ and $Al^{3+}$ share the same position, a coupled exchange of $Li^+$ and $Al^{3+}$ (0.54 Å [46]) with $Mg^{2+}$ is possible. According to Reference [51], the substitution of Al + Li = 2 Mg (a coupled exchange) between the phlogopite-trilithionite solid solutions series and a $Li^+$ exchange for the muscovite-zinnwaldite solid solutions series are documented. While this was detected at high temperatures between 500 °C and 700 °C at 2 kbar, it is ambiguous if this is performed at ambient P-T-conditions. However, higher Mg concentrations are not effective, probably due to the higher salinity and the corresponding increase, e.g., of OH groups that attach the Mg ions, so that the activity of Mg and the capability for incorporation/exchange decreases.

The experiments document that the reactions continue, and after three years, no equilibrium was reached. These reaction times are quite different to other rock types, such as basalt. For example, the reaction between basalt sand and river water seams to reach equilibrium after 126 days at 20 °C in closed experiments [52].

The experiments show that solutions with low Mg concentrations are more effective in leaching than those with higher Mg concentrations. However, the highest Li concentrations in natural brines are measured in solutions with high Mg content (Figure 7). This implies that the concentration mechanism during leaching by naturally occurring brines might have occurred later, because high Mg concentrations are less effective in leaching Li. In addition, higher temperatures and pressures favour rock–water interaction processes. Due to a maximum depth of the Zechstein evaporites of maximum 3500 m to 4000 m (Gorleben [30], Morsleben [33]) in relation to the Zechstein basis, a temperature increase of up to ca. 150 °C [38] and a lithostatic pressure of maximum 80 MPa was estimated. At present, the lithostatic pressure in Gorleben varies between 17.6 MP and 19.3 MPa at the mining level, corresponding to a rock temperature of 30–38 °C [53]. Transferring these geological conditions to the experiment, a higher Li release can be expected.

During the precipitation of salt minerals, Li salts do not form, and no Li is incorporated in the crystal lattice of naturally formed salt minerals. Despite the same charge ($1^+$), $Li^+$ (0.76 Å) does not replace $Na^+$ (1.02 Å [46]), $K^+$ (1.38 Å [46]) and $Rb^+$ (1.52 Å [46]) due to its smaller ionic radius (CN = 6). Instead, Li accumulates in evaporating seawater, with up to ca. 20 µg/g (at about 121-fold seawater concentration) during carnallite precipitation and up to ca. 26 µg/g during bischofite crystallization (Table 2, Figure 7). Therefore, if a solution yields about 19 ppm Li, it is an indicator for residual brines [14]. The appearance of brines with Li concentrations higher than about 26 µg/g thereby proves that the Li in the brines detected in the Gorleben and Morsleben mines originate from other sources. These solutions are mainly metamorphic, Mg-bearing brines. The origin/source of the Li in the brines remains ambiguous, but possible sources are:

(1) Leaching of phyllosilicate-containing strata owing to inaction with brines, e.g., postulated by References [10,28,36,54]. Following the argumentation of Reference [55], Li can be leached in minor amounts from Mg-rich, phyllosilicate-bearing strata, with Li contents of about 47 ppm to 91 ppm [56]. In fact, the maximum Li concentration of bulk rock samples in Gorleben is relatively high (330 µg/g), but lower than the maximum Li concentration detected in the brines of Gorleben (401 µg/g). While no Li analyses of the different phyllosilicate minerals in these strata exist at present, it is unclear if, or to which extent, these phyllosilicates could be responsible for the Li in the brine.

(2) Overprinting by metamorphic processes, e.g., stored in pore spaces (detected in minor quantities, maximum ca. 0.1 wt. %).

(3) Derivation from References [10,57]: organic compounds in phyllosilicate-bearing strata may contribute Li. According to Reference [2], $CaCl_2$-rich brines from coal deposits with organic compounds show higher Li concentrations. Comparable observations are documented for fluids of oil fields [23]. Beneath hydrocarbon/organic compound-bearing fluids, geothermal waters are typical Li carriers, partly characterized by elevated Li concentrations. However, this type of fluid has not been observed in the north German Zechstein salt deposits yet. At least, no resilient proof for the sources is given at present.

The experiments were created under the assumption, that leaching of phyllosilicates may be responsible for the Li in the brines to a certain degree. Therefore, the Li-endmember of the phyllosilicates (lepidolite) was used as an analogue material in order to detect leaching effects, probably in low concentrations in the interaction solution and certainly above the detection limit.

The difference between the leaching balance, detected in the reaction with NaCl solution, seems to reach saturation for lepidolite as well as, between one and three years, the most in the low concentrated $MgCl_2$ solution and K solution.

The variation in the Li concentration of the reaction solution after at least three years of reaction was relatively small, about 10 µg/g. On the other hand, the variation in the Rb content of the reaction solution of about 85 µg/g was remarkable.

## 5. Conclusions

The investigation of Li occurrences in Upper Permian (Zechstein) salt rocks and saline solutions of the Gorleben salt dome and Morsleben salt structure yield that most of the brine occurrences are connected to anhydrite rock-bearing formations and lithological boundaries. Due to their geochemical composition, especially the high Mg and Br content in relation to their mineral saturation status, this characterizes them as intrasalinar solutions, which developed during sedimentation/diagenesis and the subsequent rock–fluid interaction. Thermodynamic modelling suggests that maximum evaporated seawater shows no higher Li concentration than ca. 26 µg/g. In the Gorleben site, brines with Li concentrations of up to 401 µg/g were found, which is in line with published data for saline brines from North German salt deposits. Due to distinct higher Li concentrations in the brines, other sources of Li could be considered.

However, the origin of Li in the brines cannot be clearly defined at present. Different sources were discussed, e.g., that Li in the brines may originate from contact of saline solution with phyllosilicate-bearing strata. Other possible Li sources could be relictic metamorphic brines and organic compounds, which support enrichment of Li.

To improve the understanding of the Li origin in saline brines in an evaporitic environment, leaching experiments using a Li-bearing phyllosilicate were performed. Instead of muscovite and chlorite, which are typical phyllosilicates of the silicate-bearing strata, the more Li-containing lepidolite was used as an analogue material. Lepidolite with a Li content of 2.42 wt. % was exposed to 18 solutions of different composition (17 saline solutions and double distilled $H_2O$) for three years. The most intensive leaching effect (53.36 µg Li/g in the brine) was observed on the interaction with a 0.03 molal $MgCl_2$ solution and the second most by a modern seawater interaction (50.59 µgLi/g$_{brine}$), followed by low concentrated KCl solutions (0.1 to 0.60 molal KCl). The experiments show that Li was leached from the lepidolite, dependent on duration of the interaction reaction, the composition and pH value of the brines. Independent of these basic conditions, in all interaction solutions, with the exception of double distilled $H_2O$, lepidolite was leached, resulting in minimal contents of ca. 40 µgLi/g$_{brine}$. The experiments show that the reaction progress was not finished after three years. Additional elements were leached (e.g., Si, Rb and Cs), and composition-dependent, pH changes were detected as well.

The results from the experiments, regarding principal rock–water interaction processes, are basically transferable to the natural occurrences of phyllosilicate-bearing strata of the Gorleben and Morsleben salt deposits, but do not provide resilient proof for the origin of the Li in the investigated brines. Li-bearing phyllosilicates like muscovite and chlorites are typical in the rock types [28] and are comparable to lepidolite, which certainly shows a higher Li concentration. The Mg-containing solutions are the most effective for Li leaching by trend, which was observed in the experiments as well as in the natural brines. These Mg- and Li-enriched natural brines occurred mainly in phyllosilicate-bearing strata and anhydrite rocks or migrated through them.

The experiments lasted three years at a temperature of 22–25 °C, and it can be assumed that in geological times, much more Li might be leached due to much more reaction time and higher temperatures (maximum 150 °C, [38]). The maximum depth of the salt deposit of Gorleben and Morsleben, regarding the Zechstein basis, was 3500 m to 4000 m. The present lithostatic pressure in Gorleben is 17.6–19.3 MPa at the mining level, and the rock temperature is 30–38 °C [53].

Further investigations will be extended to Li-bearing phyllosilicates that are typical for the phyllosilicate-bearing strata, e.g., muscovite and chlorite.

To estimate the source of Li in the brines of the salt deposits and to study brine–rock interaction processes, stable isotopic investigations will be performed. The $\delta^7Li$ of seawater is ca. +31‰ and considerably higher compared to most other rock types [58]. Therefore, it should be possible to distinguish the marine origin of the brine, e.g., relictic Permian pore solutions, or freshwater. Further, the interaction between brine and allochthones-originated phyllosilicates could be determined.

**Author Contributions:** The general conceptualization for this research was by M.S. and M.M. M.S. and M.M. did the writing and figure preparation. Calculations and thermodynamic modelling were performed by M.S. Microscopy and SEM were carried out by M.M. and M.S. The experiments, accompanied sampling, including preparation, density and electrical conductivity measurements, were performed by M.S. and M.M. XRD, XRF, ICP-OES and ICP-MS analyses were realised by BGR Labs and MAS Analytics.

**Funding:** This research was funded by the Federal Ministry for Economic Affairs and Energy (BMWi), Germany, grant 9Y3215020000.

**Acknowledgments:** Many thanks to the very helpful comments and considerations of three anonymous reviewers, the editor and Linda Godfrey, who significantly helped to improve the quality of the present paper. Special thanks to our BGR colleagues Maik Gern and Tobias Faust who prepared the lepidolite, and to our analytical team Anna Degtjarev, Marina Linnenschmidt, Ragna Bergmann, Juergen Rausch and Elke Wargenau. Additional special thanks to MAS Analytics (Jens Walter and Nicol Nolte) for the geochemical analyses of the lepidolites with ICP-OES and ICP-MS. Also, thanks to Lothar Fleckenstein, Joachim Kutowski, Hartmut Blanke and Mario Patzschke (all BGE) for the assistance in the Gorleben and Morsleben mines and providing the brine samples.

Supervision and general support was from Jörg Hammer, until he passed away. Final English editing was by Alja Saßnowski.

**Conflicts of Interest:** The authors declare that they have no conflict of interest. The funders had no role in the design of the study; in the collection, analyses, or interpretation of data; in the writing of the manuscript, or in the decision to publish the results.

## Appendix A

**Table A1.** Mg and Li concentrations of the Gorleben brines (Mg in wt. % and Li in µg/g).

| Sample No. | $Mg^{2+}$ | $Li^+$ | Sample No | $Mg^{2+}$ | $Li^+$ | Sample No. | $Mg^{2+}$ | $Li^+$ | Sample No. | $Mg^{2+}$ | $Li^+$ |
|---|---|---|---|---|---|---|---|---|---|---|---|
| Go1 | 9.06 | 40.19 | Go46 | 9.05 | 35.08 | Go91 | 7.09 | 20.00 | Go136 | 7.24 | 193.00 |
| Go2 | 9.01 | 40.69 | Go47 | 9.06 | 42.75 | Go92 | 7.82 | 144.26 | Go137 | 7.08 | 170.03 |
| Go3 | 7.83 | 38.76 | Go48 | 8.85 | 34.43 | Go93 | 7.75 | 127.10 | Go138 | 0.34 | 0.36 |
| Go4 | 8.54 | 45.31 | Go49 | 9.01 | 40.69 | Go94 | 0.22 | 0.39 | Go139 | 0.18 | 1.00 |
| Go5 | 8.84 | 46.46 | Go50 | 9.06 | 40.19 | Go95 | 0.25 | 0.39 | Go140 | 0.92 | 2.00 |
| Go6 | 8.67 | 44.79 | Go51 | 8.68 | 34.32 | Go96 | 0.26 | 0.41 | Go141 | 0.88 | 2.00 |
| Go7 | 7.41 | 212.40 | Go52 | 8.86 | 40.91 | Go97 | 0.24 | 0.41 | Go142 | 0.75 | 2.00 |
| Go8 | 8.62 | 296.19 | Go53 | 8.96 | 38.78 | Go98 | 0.27 | 0.45 | Go143 | 0.75 | 2.00 |
| Go9 | 8.82 | 207.02 | Go54 | 8.86 | 39.85 | Go99 | 0.26 | 0.41 | Go144 | 0.99 | 2.00 |
| Go10 | 6.90 | 183.69 | Go55 | 8.84 | 40.41 | Go100 | 0.28 | 0.48 | Go145 | 0.99 | 2.00 |
| Go11 | 7.95 | 227.57 | Go56 | 8.98 | 39.94 | Go101 | 0.26 | 0.41 | Go146 | 0.91 | 3.00 |
| Go12 | 8.62 | 221.19 | Go57 | 8.84 | 40.26 | Go102 | 0.29 | 0.43 | Go147 | 0.74 | 3.00 |
| Go13 | 0.73 | 4.18 | Go58 | 9.41 | 78.01 | Go103 | 0.25 | 0.41 | Go148 | 0.31 | 1.00 |
| Go14 | 1.18 | 6.81 | Go59 | 8.91 | 41.57 | Go104 | 0.23 | 0.41 | Go149 | 0.42 | 3.00 |
| Go15 | 1.82 | 9.06 | Go60 | 8.92 | 43.96 | Go105 | 0.19 | 0.49 | Go150 | 0.13 | 2.24 |
| Go16 | 0.94 | 5.54 | Go61 | 8.75 | 34.36 | Go106 | 0.42 | 0.70 | Go151 | 0.39 | 2.89 |
| Go17 | 0.77 | 4.18 | Go62 | 8.81 | 79.26 | Go107 | 0.27 | 0.67 | Go152 | 0.11 | 2.49 |
| Go18 | 0.58 | 3.29 | Go63 | 9.30 | 43.80 | Go108 | 0.14 | 0.26 | Go153 | 0.13 | 2.89 |
| Go19 | 7.35 | 149.38 | Go64 | 8.88 | 42.90 | Go109 | 0.16 | 0.34 | Go154 | 0.21 | 1.98 |
| Go20 | 6.97 | 156.54 | Go65 | 6.04 | 17.00 | Go110 | 0.17 | 0.41 | Go155 | 0.07 | 2.24 |
| Go21 | 7.81 | 199.93 | Go66 | 6.12 | 32.81 | Go111 | 0.41 | 0.41 | Go156 | 0.13 | 2.24 |
| Go22 | 5.08 | 121.14 | Go67 | 6.61 | 15.00 | Go112 | 0.18 | 0.38 | Go157 | 0.12 | 1.66 |
| Go23 | 7.14 | 161.30 | Go68 | 5.79 | 17.97 | Go113 | 0.17 | 0.32 | Go158 | 0.14 | 2.57 |
| Go24 | 3.50 | 23.31 | Go69 | 8.28 | 143.00 | Go114 | 0.20 | 0.37 | Go159 | 0.15 | 1.41 |
| Go25 | 3.51 | 25.60 | Go70 | 8.20 | 156.28 | Go115 | 0.34 | 0.45 | Go160 | 0.22 | 1.57 |
| Go26 | 4.10 | 28.61 | Go71 | 8.16 | 157.72 | Go116 | 0.20 | 0.32 | Go161 | 0.14 | 2.16 |
| Go27 | 8.70 | 25.24 | Go72 | 9.12 | 40.46 | Go117 | 0.43 | 0.41 | Go162 | 0.20 | 2.16 |
| Go28 | 8.66 | 27.62 | Go73 | 9.04 | 36.66 | Go118 | 0.40 | 0.41 | Go163 | 0.11 | 1.98 |
| Go29 | 8.55 | 43.61 | Go74 | 8.99 | 39.39 | Go119 | 0.22 | 0.41 | Go164 | 0.07 | 1.33 |
| Go30 | 8.52 | 154.59 | Go75 | 7.62 | 50.30 | Go120 | 0.13 | 0.35 | Go165 | 0.08 | 1.33 |
| Go31 | 8.25 | 149.86 | Go76 | 8.29 | 218.51 | Go121 | 0.18 | 0.38 | Go166 | 0.18 | 2.72 |
| Go32 | 8.21 | 164.76 | Go77 | 0.16 | 1.24 | Go122 | 0.17 | 0.32 | Go167 | 1.12 | 2.00 |
| Go33 | 8.45 | 214.43 | Go78 | 9.00 | 22.48 | Go123 | 0.23 | 0.40 | Go168 | 1.04 | 2.52 |
| Go34 | 8.38 | 145.16 | Go79 | 8.95 | 34.71 | Go124 | 0.14 | 0.27 | Go169 | 1.07 | 3.00 |
| Go35 | 8.38 | 126.53 | Go80 | 8.53 | 32.99 | Go125 | 0.17 | 0.41 | Go170 | 1.05 | 2.00 |
| Go36 | 8.00 | 144.40 | Go81 | 8.16 | 142.20 | Go126 | 0.32 | 0.41 | Go171 | 1.12 | 2.00 |
| Go37 | 7.66 | 109.04 | Go82 | 8.14 | 162.00 | Go127 | 6.27 | 134.00 | Go172 | 1.07 | 2.00 |
| Go38 | 7.44 | 108.08 | Go83 | 8.02 | 112.04 | Go128 | 6.90 | 156.00 | Go173 | 1.12 | 2.00 |
| Go39 | 7.48 | 121.74 | Go84 | 7.96 | 173.05 | Go129 | 6.77 | 155.88 | Go174 | 1.14 | 2.00 |
| Go40 | 7.56 | 103.59 | Go85 | 6.96 | 221.29 | Go130 | 6.72 | 147.29 | Go175 | 1.11 | 2.00 |
| Go41 | 7.56 | 104.90 | Go86 | 7.37 | 193.98 | Go131 | 7.49 | 401.00 | Go176 | 1.08 | 2.00 |
| Go42 | 7.58 | 108.92 | Go87 | 7.51 | 205.38 | Go132 | 7.48 | 248.88 | Go177 | 1.09 | 3.00 |
| Go43 | 7.88 | 125.79 | Go88 | 8.64 | 307.73 | Go133 | 7.37 | 229.00 | Go178 | 1.10 | 3.00 |
| Go44 | 8.46 | 173.78 | Go89 | 8.74 | 300.89 | Go134 | 0.43 | 1.00 | Go179 | 0.39 | 2.00 |
| Go45 | 7.92 | 17.51 | Go90 | 6.17 | 9.00 | Go135 | 0.63 | 1.66 | Go180 | 0.54 | 2.00 |
| Go181 | 0.93 | 1.80 | Go193 | 4.53 | 19.46 | Go205 | 4.02 | 23.63 | Go217 | 3.56 | 28.62 |
| Go182 | 1.06 | 2.36 | Go194 | 4.88 | 17.31 | Go206 | 4.33 | 34.23 | Go218 | 4.05 | 17.50 |
| Go183 | 1.05 | 2.00 | Go195 | 5.18 | 16.96 | Go207 | 3.34 | 14.32 | Go219 | 0.88 | 9.00 |
| Go184 | 1.08 | 2.00 | Go196 | 5.20 | 24.29 | Go208 | 2.58 | 15.28 | Go220 | 1.05 | 6.00 |
| Go185 | 1.12 | 2.00 | Go197 | 5.52 | 16.03 | Go209 | 3.23 | 25.72 | Go221 | 5.97 | 3.00 |
| Go186 | 0.16 | 0.24 | Go198 | 5.44 | 13.33 | Go210 | 3.71 | 32.80 | Go222 | 1.27 | 4.16 |
| Go187 | 3.10 | 32.75 | Go199 | 5.23 | 22.45 | Go211 | 2.95 | 21.05 | Go223 | 0.55 | 2.38 |
| Go188 | 3.53 | 20.45 | Go200 | 5.22 | 45.94 | Go212 | 3.75 | 27.78 | Go224 | 0.54 | 2.55 |
| Go189 | 2.82 | 14.54 | Go201 | 5.51 | 12.52 | Go213 | 3.62 | 25.18 | Go225 | 1.39 | 3.10 |
| Go190 | 2.57 | 12.47 | Go202 | 4.81 | 39.13 | Go214 | 3.70 | 22.33 | Go226 | 0.14 | 0.35 |
| Go191 | 4.74 | 18.86 | Go203 | 5.59 | 16.49 | Go215 | 3.99 | 22.99 | Go227 | 0.14 | 0.38 |
| Go192 | 4.93 | 20.84 | Go204 | 5.14 | 13.31 | Go216 | 4.10 | 21.21 | | | |

**Table A2.** Mg and Li concentrations of Morsleben groundwater (Mg in wt. % and Li in µg/g).

| Sample No. | Mg$^{2+}$ | Li$^+$ |
|---|---|---|
| MoGw1 | 0.00 | 0.01 |
| MoGw2 | 0.01 | 0.05 |
| MoGw3 | 0.00 | 0.03 |
| MoGw4 | 0.01 | 0.06 |
| MoGw5 | 0.00 | 0.01 |
| MoGw6 | 0.00 | 0.02 |
| MoGw7 | 0.00 | 0.01 |
| MoGw8 | 0.01 | 0.01 |
| MoGw9 | 0.01 | 0.68 |
| MoGw10 | 0.00 | 0.01 |
| MoGw11 | 0.00 | 0.01 |
| MoGw12 | 0.00 | 0.01 |
| MoGw13 | 0.00 | 0.03 |

**Table A3.** Mg and Li concentrations of mining claim 1a brines of Morsleben (Mg in wt. % and Li in µg/g).

| Sample No. | Mg$^{2+}$ | Li$^+$ | Sample No. | Mg$^{2+}$ | Li$^+$ | Sample No. | Mg$^{2+}$ | Li$^+$ | Sample No | Mg$^{2+}$ | Li$^+$ |
|---|---|---|---|---|---|---|---|---|---|---|---|
| Mo1a-1 | 6.97 | 12.80 | Mo1a-16 | 6.69 | 8.74 | Mo1a-31 | 6.77 | 9.97 | Mo1a-46 | 6.67 | 10.08 |
| Mo1a-2 | 6.67 | 13.30 | Mo1a-17 | 6.75 | 9.78 | Mo1a-32 | 6.67 | 9.76 | Mo1a-47 | 6.91 | 10.20 |
| Mo1a-3 | 7.85 | 16.10 | Mo1a-18 | 6.73 | 10.20 | Mo1a-33 | 6.75 | 9.86 | Mo1a-48 | 6.82 | 9.73 |
| Mo1a-4 | 7.62 | 14.30 | Mo1a-19 | 6.55 | 10.40 | Mo1a-34 | 6.70 | 10.04 | Mo1a-49 | 6.68 | 9.50 |
| Mo1a-5 | 6.83 | 10.50 | Mo1a-20 | 6.69 | 10.70 | Mo1a-35 | 6.70 | 9.79 | Mo1a-50 | 6.74 | 9.80 |
| Mo1a-6 | 6.61 | 10.40 | Mo1a-21 | 6.75 | 11.20 | Mo1a-36 | 6.76 | 8.97 | Mo1a-51 | 6.74 | 9.19 |
| Mo1a-7 | 6.70 | 10.00 | Mo1a-22 | 6.83 | 10.30 | Mo1a-37 | 6.79 | 9.91 | Mo1a-52 | 6.78 | 9.65 |
| Mo1a-8 | 6.85 | 9.82 | Mo1a-23 | 6.90 | 10.60 | Mo1a-38 | 6.84 | 10.10 | Mo1a-53 | 6.80 | 8.89 |
| Mo1a-9 | 6.78 | 11.50 | Mo1a-24 | 6.87 | 10.30 | Mo1a-39 | 6.63 | 9.89 | Mo1a-54 | 6.44 | 9.13 |
| Mo1a-10 | 7.05 | 11.00 | Mo1a-25 | 6.90 | 10.30 | Mo1a-40 | 6.73 | 10.13 | Mo1a-55 | 6.46 | 9.27 |
| Mo1a-11 | 6.94 | 11.30 | Mo1a-26 | 6.75 | 9.88 | Mo1a-41 | 6.68 | 9.10 | Mo1a-56 | 6.48 | 9.16 |
| Mo1a-12 | 7.12 | 11.80 | Mo1a-27 | 6.78 | 9.86 | Mo1a-42 | 6.76 | 9.57 | Mo1a-57 | 6.71 | 9.34 |
| Mo1a-13 | 6.88 | 12.00 | Mo1a-28 | 6.66 | 9.66 | Mo1a-43 | 6.71 | 9.37 | | | |
| Mo1a-14 | 6.83 | 11.70 | Mo1a-29 | 6.64 | 9.56 | Mo1a-44 | 6.67 | 8.02 | | | |
| Mo1a-15 | 6.80 | 11.20 | Mo1a-30 | 6.68 | 8.58 | Mo1a-45 | 6.73 | 9.93 | | | |

**Table A4.** Mg and Li concentrations of mining claim H brines of Morsleben (Mg in wt. % and Li in µg/g).

| Sample No. | Mg$^{2+}$ | Li$^+$ | Sample No. | Mg$^{2+}$ | Li$^+$ | Sample No. | Mg$^{2+}$ | Li$^+$ | Sample No | Mg$^{2+}$ | Li$^+$ |
|---|---|---|---|---|---|---|---|---|---|---|---|
| MoH-1 | 5.90 | 1.90 | MoH-13 | 5.90 | 2.25 | MoH-25 | 5.87 | 2.19 | MoH-37 | 6.17 | 2.19 |
| MoH-2 | 5.94 | 2.46 | MoH-14 | 5.88 | 2.17 | MoH-26 | 5.88 | 2.04 | MoH-38 | 6.13 | 2.27 |
| MoH-3 | 5.95 | 2.12 | MoH-15 | 5.97 | 2.17 | MoH-27 | 6.00 | 1.81 | MoH-39 | 6.05 | 2.11 |
| MoH-4 | 5.96 | 2.28 | MoH-16 | 6.00 | 2.25 | MoH-28 | 6.00 | 2.03 | MoH-40 | 5.97 | 2.14 |
| MoH-5 | 5.92 | 2.27 | MoH-17 | 6.03 | 2.21 | MoH-29 | 6.11 | 2.13 | MoH-41 | 5.99 | 2.21 |
| MoH-6 | 6.05 | 2.58 | MoH-18 | 5.98 | 2.18 | MoH-30 | 6.04 | 2.21 | MoH-42 | 5.90 | 2.10 |
| MoH-7 | 6.00 | 1.84 | MoH-19 | 6.00 | 2.23 | MoH-31 | 5.95 | 2.30 | MoH-43 | 6.01 | 2.14 |
| MoH-8 | 5.90 | 1.82 | MoH-20 | 5.97 | 2.19 | MoH-32 | 6.02 | 2.06 | MoH-44 | 6.00 | 2.14 |
| MoH-9 | 5.94 | 2.13 | MoH-21 | 6.00 | 2.31 | MoH-33 | 5.93 | 2.18 | MoH-45 | 6.12 | 2.13 |
| MoH-10 | 5.82 | 2.20 | MoH-22 | 5.97 | 2.32 | MoH-34 | 5.93 | 2.06 | MoH-46 | 5.90 | 1.97 |
| MoH-11 | 5.97 | 2.26 | MoH-23 | 5.89 | 2.26 | MoH-35 | 6.06 | 2.04 | MoH-47 | 5.90 | 2.18 |
| MoH-12 | 5.92 | 2.29 | MoH-24 | 6.04 | 2.23 | MoH-36 | 5.88 | 2.14 | MoH-48 | 5.85 | 2.18 |

**Table A5.** Mg and Li concentrations during evaporation of seawater (Mg in wt. % and Li in µg/g). The calculations were performed with EQ3/6v7.2c.

| Calculation No. | Seawater | Mg$^{2+}$ | Li$^{+}$ |
|---|---|---|---|
| evasea1 | 1000.00 | 0.13 | 0.17 |
| evasea2 | 930.46 | 0.14 | 0.18 |
| evasea3 | 652.27 | 0.20 | 0.26 |
| evasea4 | 582.73 | 0.22 | 0.29 |
| evasea5 | 513.18 | 0.25 | 0.33 |
| evasea6 | 443.63 | 0.29 | 0.38 |
| evasea7 | 374.09 | 0.34 | 0.45 |
| evasea8 | 310.01 | 0.41 | 0.55 |
| evasea9 | 304.50 | 0.42 | 0.56 |
| evasea10 | 234.41 | 0.54 | 0.73 |
| evasea11 | 169.88 | 0.75 | 1.00 |
| evasea12 | 168.10 | 0.75 | 1.01 |
| evasea13 | 164.19 | 0.77 | 1.04 |
| evasea14 | 136.25 | 0.92 | 1.25 |
| evasea15 | 123.90 | 1.02 | 1.37 |
| evasea16 | 84.17 | 1.50 | 2.02 |
| evasea17 | 37.22 | 3.38 | 4.57 |
| evasea18 | 17.64 | 6.56 | 9.63 |
| evasea19 | 14.80 | 7.43 | 11.49 |
| evasea20 | 12.83 | 7.82 | 13.25 |
| evasea21 | 11.38 | 7.82 | 14.94 |
| evasea22 | 11.31 | 7.84 | 15.04 |
| evasea23 | 8.22 | 7.84 | 20.67 |
| evasea24 | 6.51 | 8.98 | 26.12 |

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
