# Peer review of "Lithium Occurrences in Brines from Two German Salt Deposits (Upper Permian) and First Results of Leaching Experiments"

_minerals, doi:10.3390/min9120766_

Round 1

Reviewer 1 Report

This paper investigates the origin of Li in brines associated with Zechstein salts in N. Germany. The authors sampled natural brines from the region and found concentrations up to 400 µg/g Li. They assumed that the Li comes from sediment (clay) leaching, so they performed a 3 year leaching experiment with a Li-end member phyllosilicate (lepidolite) and a variety of salt solutions (at ambient T, P). Results showed that a Mg-rich salt solution leached ~50 µg/g Li. From these data the authors propose that the source of Li is leaching of clays.

I find no rationale behind this interpretation and therefore must recommend that this paper is not ready for publication. I do not wish to discourage these authors, since they have devoted 3 years to collection of the data, however, the experimental design is flawed. The minerals in the natural system that generated the 400 µg/g Li brine are not lepidolite, but rather muscovite and chlorite. No measurement of the abundance of Li in the natural minerals was presented. I would guess it is <50µg/g, as for most such clays. No evidence is presented for dissolution of these natural minerals in the host rock which might lend credibility to the assumption that the Li came from the clays. Simple calculations could be made to show that these minerals would not account for the high Li content of the natural brines without many pore volumes of water leaching through the rock over perhaps millions of years.

The experiments are not helpful given that even 3 years of leaching under ambient conditions could not approach equilibrium. The role of pH is critical, but no discussion of it was given for the natural high Li fluids. Furthermore, the source of the Li might not be at ambient conditions, but is likely at the higher T, P of the buried rock, where cation exchange and dissolution might be in play. One could determine this by simply measuring the Li-isotopic composition of the water and comparing it to the Li-isotopic composition of the clay minerals (suspected mineral source). Are they in equilibrium at ambient conditions? Probably not, meaning the surficial rock is not the source.

One value that could be obtained from the experimental data is the solubility of lepidolite in different salt solutions. One might explore why the different solutions act differently as solvents. Since Mg and Li are similar in hydrated radius, it is interesting that the Mg salt was best at exchanging with Li. As for the clay mineralogy, the authors need to consider (and discuss) the mechanism for leaching Li out of muscovite or chlorite. The data showing evaporative increases in Li content does not approach the content measured so it is unlikely to be related to the Li source (although could be a factor).

In order to interpret the source of Li in the natural salt brines it would be more useful to study the Li-isotope composition of the fluid. It was mentioned that hydrocarbons are associated with the salt brines, and it is well known that Li is enriched in oilfield brines, so it is more likely that the high Li in the brines comes from the hydrocarbon source rock.

It is not worth a line by line critique at this point, but I suggest that the authors consider defining the word ‘bi-dest’. Is it doubly distilled water? Other English words and grammar need editing as well.

Author Response

Answers to Reviewer 1:

No measurement of Li content of natural occurring minerals were presentedà At present we have no high data density relating clay bearing strata, because it was not in the focus of our investigations yet. Therefor we started investigations also for a better understanding of the migration paths and in parts of the origin of the brines. We have no mineral separates analyzed, but bulk rock analyzes with different mineral associations of e.g. halite, anhydrite, quartz, carbonates, muscovite or chlorites with Li up to 330 µg/g. Suggesting a muscovite content of ca. 10 wt.% (of the bulk rock) results in ca. 1300 µg/g Li in the muscovite, if no other Li bearing phyllosilicate is in the sample. With higher muscovite concentrations, e.g. 20 wt.% and supposing that it is the only Li-bearing mineral in the sample, we would get a Li concentration of about 2600 µg/g. We do not agree with the reviewer’s remark in this point. Probable Li containing pore waters à We did not analyzed the water content of the salt clays. To a certain extent, some Li could be generated from pore solution. We detected up to ca. 400 µg Li/g in the brines. Supposing that all Li is stored in the pores of the clay bearing strata (no Li mineral in the sample) and suggesting 100 % pores in the clay, the Li content in the pore solution is 400 µg/g. 10% pores and achieving 400 µg/g in the brine needs a Li concentration in the pore solution of 4000 µg/g. The maximum we ever detected in salt formations was about 5 % pore water (and the rocks looked much unconsolidated). If you calculate with this untypical but in one location (not Permian) observed water content, and if you try to achieve 400 µg Li in the brine, the Li concentration in the pore solution must be 8000 µg/g. If we suggest ca. 1000 µg/g in the pores solution and the quantity of pores are 1 wt.%, you can generate 10 µg/g in the brine. These are only rough calculations, but they demonstrate witch influence different assumption have. We agree with the remark of reviewer 1 in parts, to say that Li concentrations in pores may be present, but to our opinion only with low relevance to the entire Li content. Hydrocarbons as potential source for Li à We agree in principle with this remark of reviewer 1. But hydrocarbon occurrences and their concentrations in the salt deposits of Gorleben and Morsleben are very low. But it is documented that organic compounds were observed in the clay bearing strata, but not in every sample. We implement this remark. Bi-dest. H2O à Yes, it must be double or doubly distilled H2O -> done

Using Li isotopes for detecting the origin of the Li content in the brines à Yes we envisaged this, but we are in the beginning of this project.

Reviewer 2 Report

A full review was completed with multiple comments in the pdf manuscript and some here:

In general, a interesting topic with significance and merit. I feel that the geology was neglected  (see comments below) and that more mineralogical work could have been done to support study on the brines (also authors conclude that they will do this in the future).

Comments:

Geology: The Geological Setting is covered in two sentences (l.78-82) with one (good) figure. I miss a better geological introduction especially because the reader will wonder why lepidolite was chosen and if that is because of the geology? There is mention of "Permian salt clays", but no explanation what these are and where they are found and what the geological connection is. Later in the text, there is talk of muscovite and chlorite as Li-bearer, which is out of context.

The introduction does a poor job explaining why the experiments were performed with lepidolite, only in the Conclusion it is explained clearly. I would strongly suggest to improve that.

General: use same tense throughout the manuscript, currently past and present are mixed up

good luck!

best regards

Author Response

Answers to Reviewer 2:

Not much information to the geology à We added more geological background and give more information about the clay bearing starta. Permian salt clays à We implemented a description. Link between lepidolite and chlorite, muscovite à done Explanation of the experiments in the introduction à done

All corrections are displayed in the correction modus of MS WORD according to the remarks of reviewer One. Remark of reviewer 2, calculating of the lepidolite formula, based on our data is unfortunately not possible.

XRF and ICP-MS measurements were performed, but F cannot be analyzed adequate with XRF. OH was not analyzed. Fluor analyzes have been performed only on solutions of the third year of reaction because this method was established during this time. We have not displayed these data because they are incomplete. We detected a maximum value of about 95 mg/l F in sample 41.

Reviewer 3 Report

this is an interesting paper but i have a fundamental problem with it. Firstly the authors have not considered sea water or geothermal water or even formation waters as a source of lithium yet this is more likely than a pegmatite mineral formed at high temperature- what evidence is there for pegmatites in the area?

Second lepidolite is not an analogue for clay minerals, it contains far more Li than clays and is more ordered. Surely hectorite would have been more appropriate or at least look at a range of minerals not just a high Li content mineral.

Author Response

Considering seawater, geothermal water and formation water as a source for Li instead of lepidolite Done, we discuss it together with hydrocarbon fluids. Additionally seawater and evaporated seawater are discussed in lines 501-504. Why did we use lepidolite? Lepidolite with the very high Li content was an ideal material for experimental studies, because the high Li concentration rise the chance to detect Li in the reaction solution above the detection limit even though leaching is very weak. A high concentration difference between reacting solution and the Li content in the mineral supports reactions substantial. Low Li leaching resulting in values close to the detection limit would be not very helpful. We argued that lepidolite is an analog material, which might represent principle processes related to leaching and probable exchange processes, comparable to those of muscovite and chlorite.